

# Interactive effects of acacia biochar, maize hybrids, and irrigation levels on soil health and crop productivity

Zarghoona Naz[1], Audil Rashid[1] and Summera Jahan[1,2]

[1] Department of Botany, University of Gujrat, Gujrat, Punjab, Pakistan
[2] Institute of Botany, University of the Punjab, Lahore, Punjab, Pakistan

## ABSTRACT

The demand for sustainable agricultural solutions has increased because of issues like declining soil fertility from inorganic soil changes, increasing crop water demands, shifting weather patterns, and decreasing water resources. The addition of activated acacia biochar to degraded soil can significantly influence soil health by improving its moisture and nutrient retention capacity, as well as crop productivity under water-limited conditions. At present, field experiment under split-plot design was conducted to explore the suitable level of activated biochar (A0, 0 tons ha$^{-1}$; A1, 5 tons ha$^{-1}$; A2, 10 tons ha$^{-1}$;) for maize hybrids (DK-2088, YH-5427, and DK-6317) under different moisture regimes (100% ET$_C$; full irrigation (FI), 70% ET$_C$; partially deficit irrigation (PDI), and 50% ET$_C$; severely deficit irrigation (SDI)) during maize growing year 2023 from February to June. The results showed that the addition of 10 tons ha$^{-1}$ activated biochar caused a maximum improvement in soil organic matter (109%), saturation percentage (13%), and mineral profile particularly carbon (83%) and calcium (52%). Under full irrigation (FI), activated acacia biochar amendment in the soil caused an improvement in the physiological and biochemical parameters such as sugar (55%) and protein content (136%), and yield attributes of all maize hybrids, particularly DK-6317. However, under severely deficit irrigation (SDI), the highest improvement in protein content, and yield per hectare was found in DK-2088, *i.e.*, 11% to 29% higher in 5 tons ha$^{-1}$, 10 tons ha$^{-1}$ activated biochar amended soil, respectively. The average economic gain percentage was highest in DK-6317, *i.e.*, 1-fold, and 2.5-fold higher under PDI, and SDI in 10 tons ha$^{-1}$ activated biochar amended soil. The present study indicates the significance of organically activated acacia biochar amendments in soil for improving its water retention capacity and enhancing maize growth and yield under moisture-deficit conditions.

## INTRODUCTION

Under natural field conditions, crops are frequently subjected to various biotic and abiotic stresses throughout their growth and development, from seedling emergence to harvest. Among these stresses, water scarcity has emerged as one of the most critical factors adversely affecting crop productivity (*Brodersen et al., 2019*). Historical evidence highlights

Corresponding authors
Audil Rashid, a.rashid@uog.edu.pk
Summera Jahan,
summera.botany@pu.edu.pk

the impact of water shortages in triggering severe famines, thereby posing a serious threat to global food security (*Huang et al., 2022*). The enormously growing global population has intensified agricultural activities and increased crop demand, which in turn has led to the overexploitation of natural resources, resulting in declining soil fertility and reduced soil water-holding capacity. Maize (*Zea mays* L.), a widely cultivated and versatile crop across diverse agro-climatic regions, holds significant economic importance as both food and feed (*Sah, Choudhury & Das, 2016*). It is extensively used in the production of sweeteners, cooking oil, adhesives, pharmaceuticals, beverages, alcoholic products, and starch (*Ranum, Peña Rosas & Garcia-Casal, 2014*). Despite its multifaceted utility, the average maize yield in Pakistan during 2020–2021 was 6.3 tons per hectare, substantially lower than other maize producing countries, such as Turkey (11.45 tons ha$^{-1}$), the United States (10.76 tons ha$^{-1}$), and Egypt (8 tons ha$^{-1}$) (*USDA, 2022*). The primary factors contributing to this yield gap include water scarcity, elevated temperature and suboptimal crop management practices (*Yousaf et al., 2021*). Maize undergoes several critical developmental stages, including seedling establishment, vegetative growth, tasseling and maturity, all of these developmental phases are highly vulnerable to abiotic stresses (*Gul et al., 2015*).

Numerous research studies have reported that water stress adversely affects the morphological, biochemical, and physiological traits of maize, leading to yield losses ranging from 10% to 27% under water-deficit conditions (*Leng, 2021*; *Yousaf et al., 2022*). To counter these challenges, eco-friendly and sustainable agricultural practices are required (*Bruulsema, 2018*).

Among organic soil amendments, biochar stands out as a promising strategy. Biochar is a fine-grained, porous material produced through the pyrolysis of organic biomass at high temperatures under limited oxygen conditions. It has been widely recognized for its potential to enhance soil fertility, nutrient retention, and carbon sequestration, while also reducing $CO_2$ emissions (*Fazal & Bano, 2016*; *Zhang et al., 2020*). Biochar applications improve key soil properties by increasing water retention, decreasing bulk density, enhancing pore volume, and promoting soil aggregation, especially in coarse-textured soils (*Wang et al., 2019*). It also contributes essential nutrients such as potassium, magnesium, and calcium, and mitigates nutrient leaching, thereby improving crop resilience to drought (*Jahan et al., 2023*; *Li et al., 2021*). Notably, its high porosity enables it to retain capillary water within its micropore structures for extended periods, supporting plants during water scarcity (*Paetsch et al., 2018*; *Jahan et al., 2019*). Acacia wood, abundant in southeast Asia, is a most suitable feedstock for biochar production due to its availability and high lignocellulosic content (*Shakeel et al., 2022*). The use of acacia-derived biochar is particularly suitable for arid and semi-arid regions, where water shortage is intensified by climate change. Various research studies indicate that biochar-amended soil promotes microbial activity essential for organic matter decomposition and nutrient cycling, especially nitrogen and phosphorus (*Dai et al., 2021*). These effects collectively improve soil structure, water availability, and crop productivity, highlighting biochar as a "win-win" solution for drought affected agriculture (*Chen et al., 2023*).

Field crop, including maize, experience multiple stressors during their growth cycle. Water constitutes 80–90% of a plant's fresh biomass and is essential for vital

physiological processes such as photosynthesis, adenosine triphosphate (ATP) synthesis, and nutrient transport (*Abbasi & Abbasi, 2010*; *Brodersen et al., 2019*). Water stress is therefore considered one of the greatest threats to food security, and historically, it has been a leading factor in widespread famine (*Okorie, Mphambukeli & Amusan, 2019*). The current study investigates the effects of activated acacia biochar on soil health and maize productivity under drought stress. It hypothesizes that activated biochar-amended soil enhances the maize growth and yield by improving moisture retention and altering soil physicochemical properties. While global food demands are rising, overharvesting and poor crop rotation practices can degrade soil quality if not managed properly (*Lema et al., 2019*). Although various amendments, such as animal manure, organic matter, lime, gypsum are used to improve soil fertility, biochar is considered one of the most effective and environmentally friendly solutions (*Jahan et al., 2020*; *Kloss et al., 2014*).

Organically activated biochar creates a favorable microenvironment that supports microbial processes, which in turn releases essential nutrients (*Jahan et al., 2023*). These improvements lead to more efficient water use, better root growth, and enhanced nutrient uptake, ultimately supporting higher crop yields under water-deficit conditions. Despite its potential, most current research is confined to controlled environments, limiting the understanding of its performance in real-world settings. This study addresses that gap by evaluating organically activated biochar under field conditions in arid region (Gujrat) of Pakistan. It provides novel insights into the effects of biochar on different maize hybrid verities and identifies the optimal application level for improving crop productivity. The present work assesses the effectiveness of vermicompost and perlite-based activated biochar in improving maize performance in field conditions. By bridging laboratory findings with practical applications, this research work offers a sustainable approach to maintaining agricultural productivity in water-limited environments.

## MATERIALS AND METHODS

### Biochar production and activation

Biochar was produced from *Acacia nilotica* wood through slow pyrolysis at 450 °C for 3 h in a temperature-controlled pyrolysis unit, as described by *Jahan et al. (2022)*. For biochar activation vermicompost and perlite were used. Vermicompost was prepared using corn stover as the feedstock and red earthworm (*Eisenia fetida*) as the composting organisms. To activate the biochar, it was mixed with perlite and vermicompost in a 1:1:1 ratio by weight. For enhancing activation process, a 2-liter liquid molasses solution was added to the mixture, which was stirred daily for proper aeration. The activation process continued for three weeks until excess moisture had completely evaporated (*Jahan et al., 2023*).

The field experiment was carried out from February to June, 2023. The study site was located in a loamy soil region of Gujrat, (Pakistan) characterized by slightly basic pH. Prior to sowing, the soil was ploughed three times using a tractor and a rotavator to ensure proper soil tilth, followed by irrigation through field water channels. Three levels of activated biochar amendments were incorporated into the top 15 cm of soil; 0 tons ha$^{-1}$ (A0), 5 tons ha$^{-1}$ (A1), and 10 tons ha$^{-1}$ (A2). The primary objective of activated

biochar application was to enhance soil water-holding capacity and organic matter content by improving soil structure and chemical interactions (*Paetsch et al., 2018*). Soil samples were collected to analyze their physico-chemical properties.

## Soil analysis

The baseline edaphic characteristic of the field soil were as follows: 279 g kg$^{-1}$ silt, 389 g kg$^{-1}$ clay and 330 g kg$^{-1}$ sand, classifying it as loam. Structural and elemental analyses of both amended and non-amended soil were performed using Scanning Electron Microscopy coupled with Energy-Dispersive X-ray spectroscopy (SEM-EDX; JSM-5910, JEOL, Japan). Soil organic matter content was determined according to the method of *Estefan, Sommer & Ryan (2013)*. Soil pH was measured in air-dried soil samples following the ISO 10390 standard. Electrical conductivity (EC) was assessed using the procedure of *Rayment & Higginson (1992)*. Saturation percentage (Sp) was calculated using the protocol outlined by *Estefan, Sommer & Ryan (2013)*.

$$SP = \frac{\text{Total weight of H}_2\text{O}}{\text{Weight of dried soil}} \times 100.$$

## Field experiment and experimental design

The experiment was conducted using a split-split-plot design with three replications. The total experimental area was 4,640 m$^2$. Each replication was divided into main plots (72 m$^2$) and sub-sub-plot of 4 × 4 m$^2$. A two m wide path separated each sub-plot and sub-sub plot and water channels were established along these paths to facilitate controlled irrigation and prevent horizontal flow of water and cross contamination between treatments. The maize hybrids DK-2088, DK-6317, and YH-5427 were sown manually at a rate of 25 kg ha$^{-1}$, by maintaining a plant-to-plant spacing of 12.5 cm. Sowing was carried out on February 18, 2023.

## Irrigation requirements (IR) for crop

Three irrigation regimes were implemented in the field experiment; full irrigation (FI) at 100% of crop evapotranspiration (ETc), partially deficit irrigation (PDI) at 70% ETc, and severely deficit irrigation (SDI) at 50% ETc. Daily reference evapotranspiration (ET$_0$) was determined using the FAO Penman-Monteith equation (*Allen et al., 1998*). The reference evapotranspiration values (expressed in mm day$^{-1}$) along with crop coefficient (Kc) values specific to maize growth stages, were used to calculate the crop's water requirement. The following standard equation, as given by the Food and Agriculture Organization (FAO), was used to estimate irrigation needs.

$$IN = ETC - Pe$$

where Pe indicates the effective rainfall, ETc is crop evapotranspiration, IN is the net water requirement. The evapotranspiration was calculated further by using the following formula:

$$ETC = KC \times ET_0$$

where; $ET_0$ is the reference evapotranspiration, and Kc is crop coefficient for maize. The reference evapotranspiration was measured by using the Monteith equation (*Howell & Evett, 2004*).

## Assessment of plant physiological and biochemical stress biomarkers

Proline content in maize leaf was determined by the procedure of *Bates, Waldren & Teare (1973)*. Using the *Prochazkova et al. (2001)* method, the lipid peroxidation of maize leaves was recorded.

$$MDA = \frac{\left[ (A532 - A600) - \left[ (A440 - A600) \left( \frac{MA\ at\ 532}{MA\ at\ 440} \right) \right] \right]}{157000} \times 10^6.$$

MA represents the molar substance in sucrose at two wavelengths (532 nm and 600 nm, respectively).

Leaf sugar content was estimated by crushing 0.5 g of leaf material in a clean mortar with 10 mL of distilled water (*DuBois et al., 1956*). After crushing, the mixture was filtered. In a test tube, 0.1 mL of filtrate with one mL of phenol (5% v/v) was added. After incubation for one hour, at room temperature, five mL of concentrated $H_2SO_4$ was added in the test tube. The same procedure was repeated for the blank and filtrate from other treatments. The absorbance readings of the sample and blank was noted at 420 nm by using a spectrophotometer (UV-Vis; Shmadzu 35, RP China).

For antioxidant enzyme assay, maize (one g) leaves were ground in a mortar and mixed with five mL of phosphate buffer and the mixture was centrifuged at 13,000 g at 4 °C for 19 min, and the upper layer was used to test the enzyme activity. The peroxidase was assayed by the procedure of *Naveh, Mizrahi & Kopelman (1981)*.

$$POD = \frac{\Delta 485}{mg\ of\ Protein}.$$

For catalase, the reaction mixture (three mL) was prepared by mixing 0.2 mL of enzyme extract and 50 mM $H_2O_2$. After 5 min, titanium reagent was added to terminate the reaction (*Taranishi et al., 1974*). Absorbance readings were noted at 410 nm, and following formula was used to assess catalase:

$$CAT = \frac{\Delta 410}{mg\ of\ Protein}.$$

## Assessment of morphological parameters

The plant height and leaf area were determined by ImageJ software (v1.51j8, USA). The fresh and dry weights of the crop were measured at 15-day intervals. Leaf, shoot, and root fresh weights were determined by an analytical weighing balance. Later, the root, shoots, and leaves were dried inside an oven for 72 h at 60 °C, and their dry weights were determined.

## Yield parameters

The yield parameters including hundred seed weight, cob length, yield kg hectare$^{-1}$ were determined by *Parihar et al. (2018)*.

### Economic analysis of yield

An economic analysis of the maize crop was conducted using standard procedures, and average values were calculated across replications. Total production costs were estimated by counting for all inputs, including seed cost, land rent, field preparation, fertilizers, plant protection measures, and labor wages, following the approach outlined by *Gittenger (1982)*. The gross income was determined based on prevailing market price of maize grain at the time of harvest. Net profit or loss was then calculated by subtracting the total cost of production from gross income. To assess the economic feasibility of the treatments, the benefit-cost ratio (BCR) was determined by using the following formula:

$$BCR = \frac{Gross\ income}{Total\ cost}.$$

### FTIR analysis of maize grain

The Fourier-transform infrared (FTIR) spectra of dry maize seed powder were obtained using the KBr (potassium bromide) method, utilizing a Nicolet iS5 Thermo Scientific instrument (USA), following the procedure outlined by *Celi, Schnitzer & Négre (1997)*. The infra-red spectra of the dry maize seed samples were obtained in the range of 4,000 to 500 cm$^{-1}$. Further processing and analysis of the spectra were conducted using Origin Pro 2023.

### Statistical analysis

Data were statistically analyzed using a three-way analysis of variance (ANOVA) under the general linear model (GLM) framework in Minitab 17. Mean comparisons were performed using Tukey's test at a 5% significance level. All reported values represent the mean values $\pm$ standard error of the mean (SEM), based on three replicates ($n = 3$). Fourier-transform infrared (FTIR) spectral data from soil and maize seed samples were plotted using Origin Pro 2019. To explore the relationship between soil amendments, soil properties and plant responses, heatmap analysis was conducted using R statistical software (version R-4.3.1) following the approach described by *Barter & Yu (2018)*.

## RESULTS

### Physicochemical characteristics of soil

The saturation percentage (Table 1) was recorded to be considerably higher (10–13%) with A1 (5 tons ha$^{-1}$ activated biochar) and A2 (10 tons ha$^{-1}$ activated biochar) in soil than A0 (soil without activated biochar). Regarding soil pH, it was noted that activated biochar buffered the soil to attain neutral pH and it was improved by 11%–19% with A1 and A2 incorporation in soil in contrast to A0. Similarly, soil organic matter was enhanced (50%–109%) with A1 and A2 activated biochar amendments. The electrical conductance of soil was also improved (18%–30%) in A1 and A2 amended soil. The available phosphorus and potassium were considerably improved *i.e.,* 50%–125% and 8%–21% higher, respectively, in A1 and A2 incorporated soil.

**Table 1  Physicochemical analysis of the soil including saturation percentage (sp), organic matter (OM), pH, electrical conductivity (EC), available phosphorus and available Potassium content (k) in NS, A1, A2.**

| No | Parameters | NS | A1 | A2 |
|----|-----------|-----|-----|-----|
| 1 | SP (%) | 40 ± 1.15 c | 44 ± 1.7 b | 45.3 ± 0.57 a |
| 2 | OM (%) | 0.22 ± 0.01 c | 0.33 ± 0.01 b | 0.46 ± 0.01 a |
| 3 | pH | 6.3 ± 0.30 b | 7.06 ± 0.11 a | 7.3 ± 0.05 a |
| 4 | EC (dS m$^{-1}$) | 0.62 ± 0.01 c | 0.73 ± 0.01 b | 0.81 ± 0.01 a |
| 5 | Available Phosphorus (mg kg$^{-1}$) | 4.86 ± 0.96 b | 6.16 ± 0.55 b | 9.2 ± 1.03 a |
| 6 | Available Potassium (mg kg$^{-1}$) | 99.6 ± 1.52 c | 121 ± 2.08 b | 107 ± 1.15 a |

**Notes.**
Means sharing different alphabets (a–c) significantly differ at $P < 0.05$. ± Value indicates the standard deviation of three repli­cates. NS (non-amended soil), A1 (activated biochar 5 tons ha$^{-1}$), A2 (activated biochar 10 tons ha$^{-1}$).

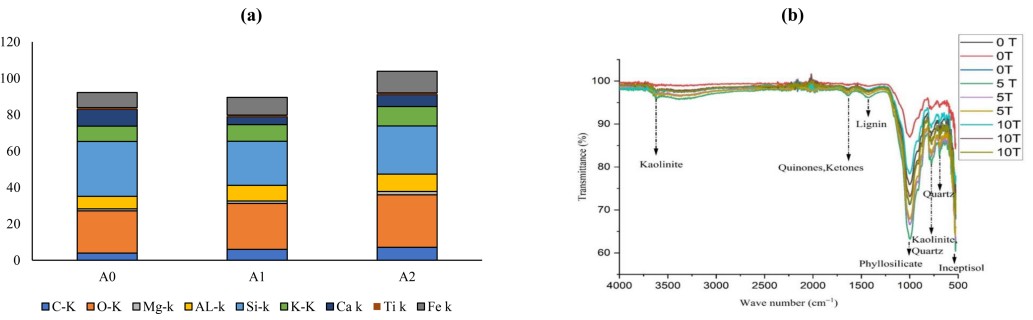

**Figure 1  Impact of activated biochar on soil (A) elemental composition of soil (B) FTIR analysis of soil 0 tons ha$^{-1}$ (A0), 5 tons ha$^{-1}$ (A1), 10 tons ha$^{-1}$ (A2) activated biochar amended soil.**

## Percent weight of elements in the soil

The surface characteristics and elemental composition of soil samples is presented in Fig. 1A and Fig. S1. The carbon percentage in soil was increased by 54% to 83% in A1 and A2 soil amended with activated biochar as compared to A0. The oxygen content was also enhanced by 10% to 43% in soil with activated biochar in contrast to non-amended soil. Magnesium content in soil was enhanced (52% to 10%) in A1 and A2 soil than in A0. The aluminum content in soil increased by 41% to 27% in A1 and A2 than in A0. Silicon in soil was lowered by 12% to 20% in A1 and A2 in contrast to control A0. The potassium percentage was increased by 26% to 9% in activated biochar-amended soil. The calcium percentage decreased by 29% to 52% in A1 and A2 soil than to non-amended soil. The titanium percentage was declined by 8% to 16% in A1 and A2 soil as compared to A0. The iron percentage was improved by 45% to 18% in A1 and A2 soil amended with activated biochar then to A0.

## Soil FTIR analysis

The FTIR spectra of the soil showed the intense bands in the activated biochar amended soil (Fig. 1B). The OH band at 3,620 cm$^{-1}$ is 32% and 6% higher in A1 and A2 then to A0 soil. While, C = C band at 1,456 cm$^{-1}$ is 18% to 13% higher in A1 and A2 then to A0.

The Si-O band at 1,006 cm$^{-1}$ is 2% to 1.4% higher intensity in A2 and A1 soil then to A0 (*Xu et al., 2020*).

## Weather conditions

The climatic data is presented in Fig. S2 for the year 2023, mean monthly temperatures ranged from 21.9 °C (February) to 36.7 °C (June), with a gradual increase over the months. Mean relative humidity varied between 30.9% (February) and 39.4% (March), while wind speeds fluctuated from 2.41 m/s (March) to 3.21 m/s (May). Average daily sun hours remained consistently high, ranging from 10.12 (February) to 11.60 (May). Total monthly rainfall was absent in February (0.00 mm), peaked in March (83 mm), and reached its highest in June (134.1 mm).

## Estimation of biochemical stress indicators

The highest increase of sugar content was found in Dk-6317 (32% to 55%) as compared to Dk-2088 and Yh-5427. While lowest sugar content was found in Dk-2088 under all irrigation levels (Fig. 2A). The proline content was higher in Dk-2088 in contrast to Yh-5427, and DK-6317. While the lowest proline content under all irrigation levels was found in Dk-6317 (Fig. 2B). The protein content was higher in Dk-2088 (17% to 136%), as compared to Yh-5427 and Dk-6317. The lowest protein content was found in Yh-5427 (70% to 62%) under all irrigation (Fig. 3A). The lipid peroxidation in Dk-2088, Yh-5427, and DK-6317 were decreased by 33% to 91%, 34% to 61% and 38% to 70.4% in A1 and A2 soil as compared to A0 under full irrigation. The lowest lipid peroxidation was detected in Dk-6317 by 25% to 32% in A1 and A2 soil when compare to A0 under severely deficit irrigation conditions (Fig. 3B). Considerably enhanced leaf peroxidase (POD) activity was recorded in the plants of non-amended soil; however, comparatively low POD activity was recorded in the plants of activated biochar-amended soil. The POD activity reduced under water stress in Dk-2088, Yh-5427, and DK-6317 by 36% to 66%, 41% to 60%, and 57% to 64% in A1 and A2 then to A0 (Fig. 3C). The catalase activity enhanced considerably in non-amended soil. However, the reduction of catalase activity (116% to 84%) was found in Yh-5427 in contrast to Dk-2088 and Dk-6317 under full irrigation in A1 and A2 soil then to A0. The lowest catalase activity was found in Dk-6317 (−29% to −48%) (Fig. 3D).

## Morphological parameters

The highest increase in shoot fresh weight (Table S1) was observed in Dk-6317 at the vegetative and maturity stages, with improvements ranging from 4% to 19% and 4% to 8%, while at the tasseling stage, shoot fresh weight was improved considerably in Yh-5427 by 4% to 8% respectively, under full irrigation in A1 and A2 soil. Under partially deficit irrigation conditions, improvement in shoot fresh weight was recorded in Dk-6317 at the vegetative, tasseling, and maturity stages, with enhancements ranging from 6% to 37%, 12% to 13%, 5.5% to 8% respectively, while the highest shoot fresh weight under severely deficit irrigation conditions was recorded in DK-6317 at the vegetative, tasseling and maturity stages *i.e.,* 38% to 58%, 6% to 9%, and 4% to 11% respectively in treatments A1 and A2. Conversely, the greatest improvement in shoot dry weight was noted in Dk-6317 at the vegetative, tasseling, and maturity stages, with enhancements ranging from 99% to
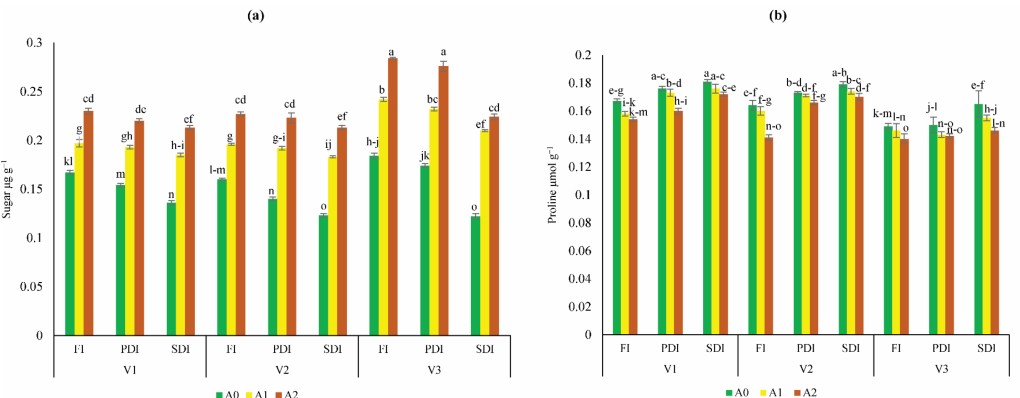

**Figure 2** **Impact of activated biochar amendment in soil on three maize hybrids on (A) sugar content (B) proline content under full irrigation (FI), partially deficit irrigation (PDI), and severely deficit irrigation (SDI).** Bars are indicating mean values of three replicates ($n = 3$). Means sharing different alphabets significantly differ at $P < 0.05$. Error bars are indicating the standard deviation of three replicates. NS (non-amended soil), A1 (activated biochar 5 tons ha$^{-1}$), A2 (activated biochar 10 tons ha$^{-1}$). V1 (DK-2088), V2 (Yh-5427), V3 (DK-6317).

161%, 5% to 21%, 36% to 55% respectively, under severely deficit irrigation conditions in treatments A1 and A2.

Moreover, the most notable enhancement in root fresh weight was observed in Dk-6317, *i.e.,* ranging from 107% to 153% at the vegetative stage (Table S2). At the tasseling and maturity stages, the most significant improvements in root fresh weight were observed in Dk-6317 *i.e.,* 54% to 150%, 42% to 155% under severely deficit conditions in treatments A1 and A2 compared to treatment A0. Under severely deficit irrigation, the highest increase in root dry weight at the vegetative stage, tasseling, maturity stages 79% to 208% and 7% to 80% in A1 and A2 then to A0.

In terms of leaf fresh weight (Table S3), the highest improvement at the vegetative stage was found in Yh-5427 ranging from 15% to 89%, while at tasseling and maturity stages, improvement was detected in Dk-6317, *i.e.,* ranging from and 21% to 87% and 15% to 37% respectively, under severely deficit irrigation in treatments A1 and A2 compared to treatment A0. Under severely deficit irrigation, the highest increase in leaf dry weight at the vegetative stage was found in Dk-6317 *i.e.,* 128% to 220%. Regarding plant height, the highest improvement at the vegetative stage was observed in DK-2088 by 44% to 47%. At the tasseling and maturity stages, the highest improvements were found in YH-5427 by 6% to 21% and 30% to 35% in treatments A1 and A2 compared to A0 under partially deficit irrigation.

Regarding plant height (Table S4) at the vegetative stage, the highest improvement was observed in DK-2088 by 43% to 47%. At the tasseling and maturity stages, the highest improvements were found in YH-5427 by 5% to 21% and 30% to 35% in treatments A1 and A2 compared to A0 under partially deficit irrigation. Across all growth stages, the most substantial increase in leaf area under partially deficit irrigation was observed in YH-5427 by 19% to 42% in treatments A1 and A2 compared to A0 (Table S4). At the vegetative and

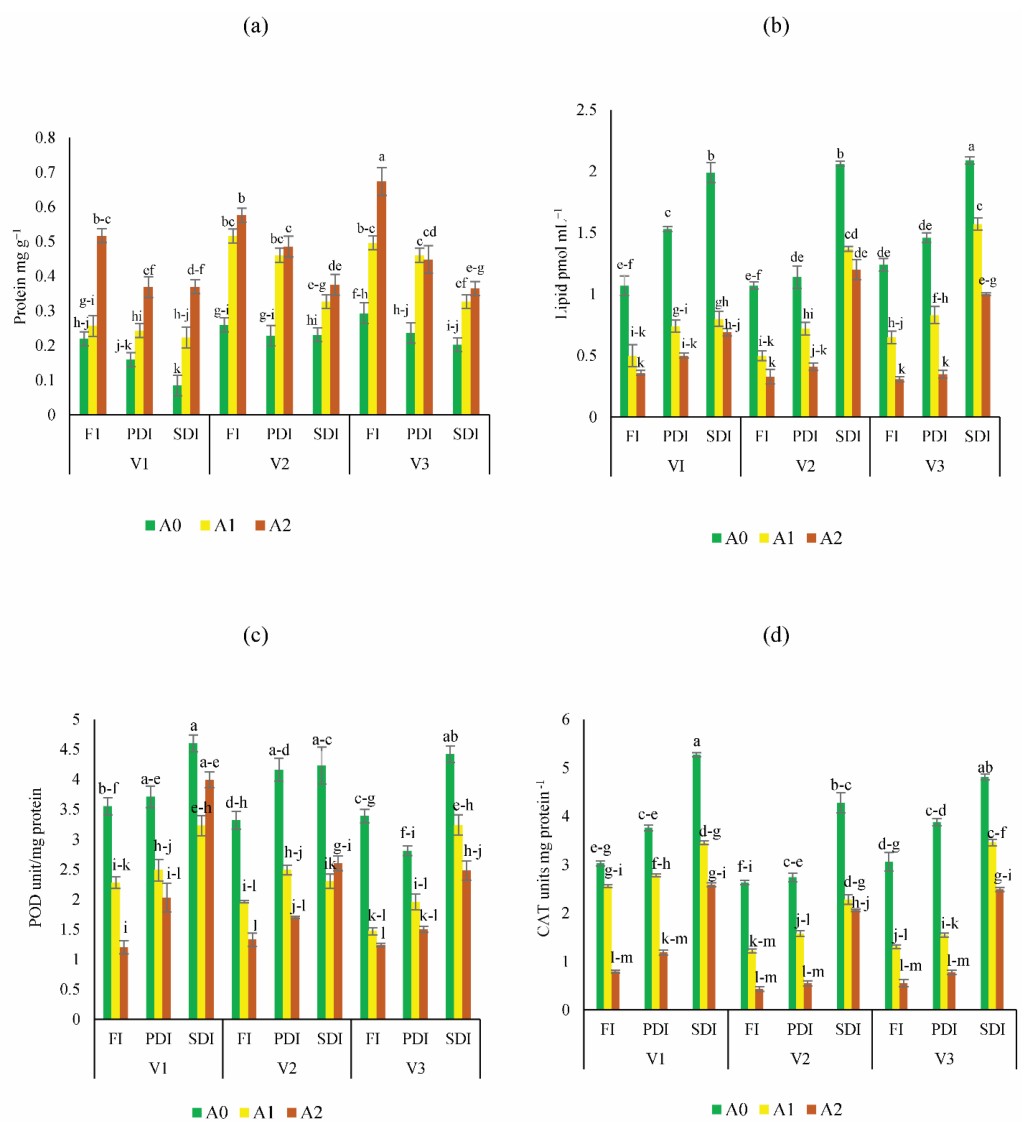

**Figure 3 Impact of activated biochar amendment in soil on three maize hybrids on (A) protein content (B) lipid peroxidation (C) peroxidase activity (POD) (D) catalase (CAT) activity.** Bars are indicating mean values of three replicates ($n = 3$). Means sharing different alphabets significantly differ at $P < 0.05$. Error bars are indicating the standard deviation of three replicates. NS (non-amended soil), A1 (activated biochar 5 tons ha$^{-1}$), A2 (activated biochar 10 tons ha$^{-1}$). V1 (DK-2088), V2 (Yh-5427), V3 (DK-6317). Full irrigation (FI), partially deficit irrigation (PDI), and severely deficit irrigation (SDI).

maturity stages, the highest increase in leaf area was found in Dk-6317 by 28% to 37% and 8% to 11%, respectively, while at the tasseling stage, highest improvement was observed in YH-5427 by 19% to 44% under severely deficit irrigation in treatments A1 and A2 compared to A0. Regarding grain fresh weight, the most substantial increase was observed in Dk-2088 (45% to 130%) under full irrigation in treatments A1 and A2 compared to treatment A0 (Table S4).

## Yield attributes

After experiencing water stress, plants grown in non-amended soil exhibited significantly lower yields (Fig. 4). However, activated biochar-amended soil demonstrated notable enhancements in the hundred seed weight of Dk-2088, Yh-5427, and Dk-6317, ranging from 5% to 49%, 9 to 55%, and 51% to 65% respectively, in A1 and A2 respectively, compared to A0 under partially deficit irrigation. Similarly, the hundred seed weight also improved considerably in activated biochar amended soil under severely deficit irrigation, with Dk-2088 exhibiting an increase of 29% to 52%, Yh-5427 showing an increase of 33% to 67%, and Dk-6317 displaying an increase of 31% to 45% in treatments A1 and A2 compared to A0 (Fig. 4A). The yield per hectare of maize was significantly influenced by various doses of activated biochar (Fig. 4B). Application of activated biochar enhanced yield per hectare in Dk-2088 by 5% to 15%, in Yh-5427 by 13% to 27%, and in Dk-6317 by 28% to 33% in A1 and A2 compared to A0 under full irrigation. Furthermore, under partially deficit irrigation, the yield per hectare of Dk-2088 increased by 9% to 24%, in Yh-5427 by 18.4% to 27%, and in Dk-6317 by 26% to 34% when soil treated with A1 and A2 compared to A0. Similarly, yield per hectare under severely deficit irrigation increased in Dk-2088 by 11% to 29%, in Yh-5427 by 17% to 14% and in Dk-6317 by 11% to 24% in soil amended with activated biochar (A1 and A2) compared to non-amended soil. The highest increase in cob length (Fig. 4C) was observed in Dk-2088, ranging from 8% to 27%, compared to Yh-5427 and Dk-6317. Regarding stover yield, notable enhancements were observed in Dk-2088, Yh-5427, and Dk-6317, ranging from 15% to 34%, 29% to 36% and 8% to 59%, respectively, under full irrigation in A1 and A2 compared to A0. Conversely, under partially deficit irrigation, stover yield (Fig. 4D) increased by 7% to 38% in Dk-2088, by 22% to 28% in Yh-5427, and by 22% to 40% in Dk-6317 when comparing treatments A1 and A2 to A0. Under severely deficit irrigation conditions, stover yield increased by 30% to 58% in Dk-2088, by 18% to 33% in Yh-5427, and by 47% to 65% in Dk-6317 in soil treated with activated biochar (treatments A1 and A2).

## Heatmap analysis of interaction of soil properties and plant growth characteristics

A heatmap analysis (Fig. 5) was conducted to examine the interactions among different levels of activated biochar applications *i.e.,* 0 tons ha$^{-1}$(A0), 5 tons ha$^{-1}$ (A1), 10 tons ha$^{-1}$ (A2), across three maize hybrids, DK-2088 (V1), Yh-5427 (V2), DK-6317 (V3) under three moisture level full irrigation (FI), partially deficit irrigation (PDI), and severely deficit irrigation (SDI). The results revealed that the A2 treatment under FI significantly improved soil organic matter and mineral content, and plant physiological and biochemical parameters across all three hybrids (A2+FI+V1, A2+FI+V2, and A2+FI+V3). In contrast, A0 under SDI showed marked reductions in soil health and plant performance (A0+SDI+V1, A0+SDI+V2, A0+SDI+V3). Notably, both activated biochar levels (A1 and A2) improved soil quality and plant responses under stress conditions, with the A2 offering the most substantial benefits even under SDI (A2+SDI+V1, A2+SDI+V2, A2+SDI+V3).

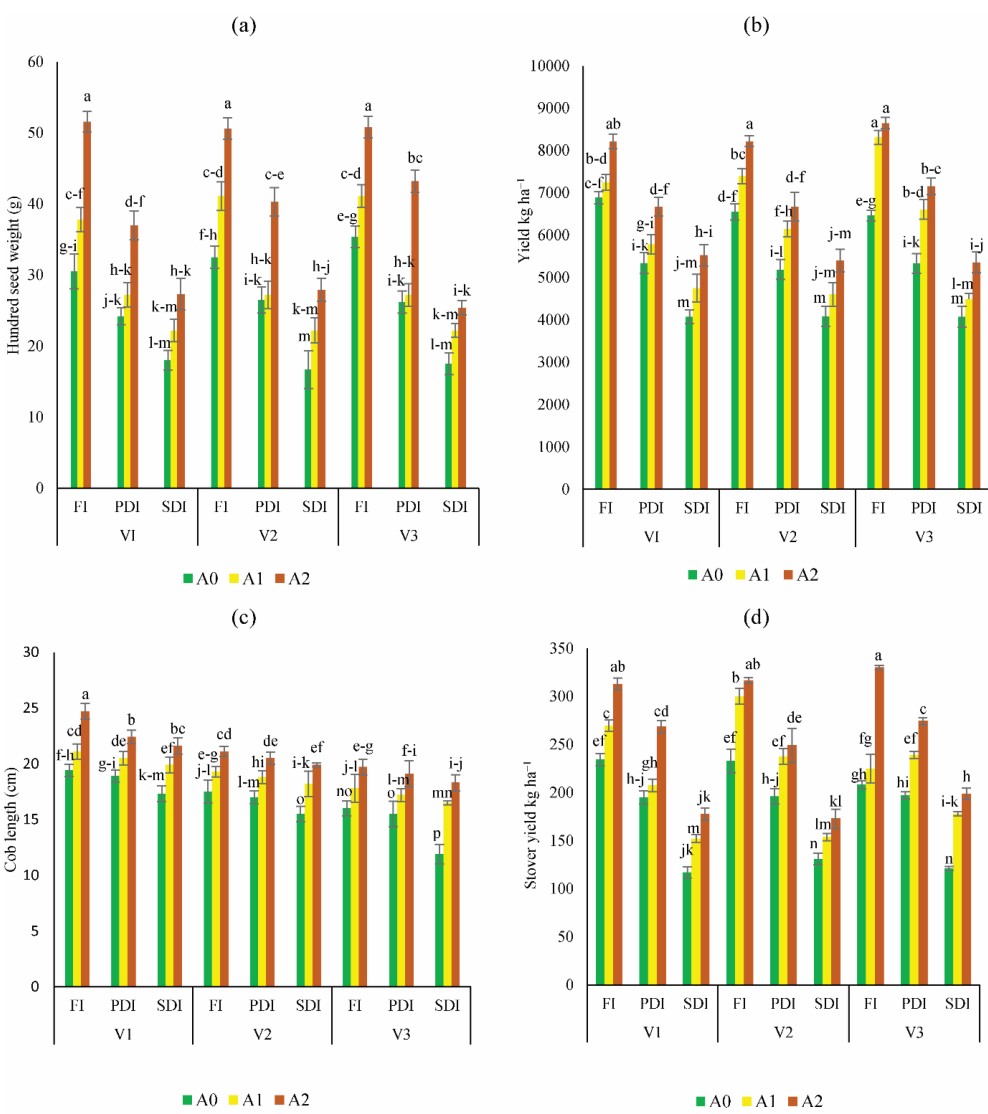

**Figure 4  Impact of activated biochar amendment in soil on three maize hybrids on (A) hundred seed weight (B) yield kg ha$^{-1}$ (C) cob length (cm) (D) stover yield (kg ha$^{-1}$).** Bars are indicating mean values of three replicates ($n = 3$). Means sharing different alphabets significantly differ at $P < 0.05$. Error bars are indicating the standard deviation of three replicates. NS (non-amended soil), A1 (activated biochar 5 tons ha$^{-1}$), A2 (activated biochar 10 tons ha$^{-1}$). V1 (DK-2088), V2 (Yh-5427), V3 (DK-6317). FI (Full irrigation), PDI (partially deficit irrigation), and SDI (severely deficit irrigation).

## Economic analysis of maize

The average net returns from maize cultivation varied significantly among the three hybrids and irrigation regimes, depending on the level of activated biochar application (Table S5). For Dk-2088, net returns ranged from 1,303 to 1,526 USD under full irrigation, 796 to 1,087 USD under partially deficit irrigation, and 463 to 637 USD under severely deficit irrigation. For Yh-5427, returns ranged from 1,323 to 1,628 USD under FI, from 908 to 1,065 USD under PDI and from 402 to 603 USD under SDI. Meanwhile, Dk-6317 exhibited

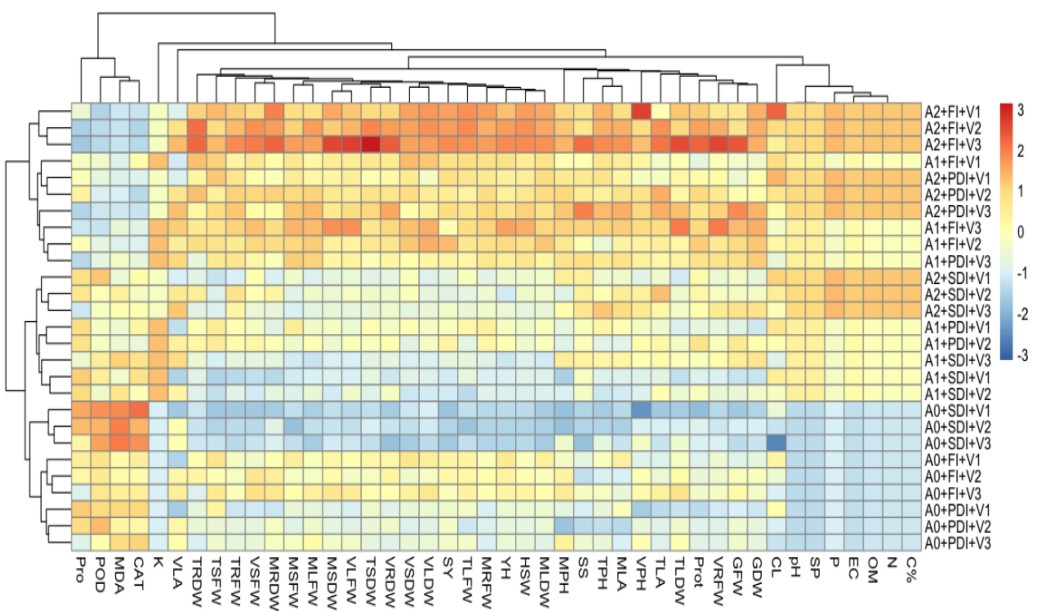

**Figure 5  Heatmap showing correlation between amendment levels of activated biochar 0 tons ha$^{-1}$ (A0), 5 tons ha$^{-1}$ (A1), 10 tons ha$^{-1}$ (A2) on physiological, biochemical, growth and yield attributes of maize.** V1 (DK-2088), V2 (Yh-5427), V3 (DK-6317) on C% (carbon percentage), N (nitrogen percentage), OM (organic matter), EC (electrical conductivity), P (phosphorus content), SP (saturation percentage), pH (potential of hydrogen) of soil and CL (cob length), MPH (plant height at maturity stage), VPH (plant height at vegetative stage), MRDW (root dry weight at maturity), VSFW (shoot fresh weight at vegetative stage), TLDW (leaf dry weight at tasseling), VRDW (root dry weight at vegetative stage), TSDW (shoot dry weight at tasseling), SY (stover yield), TLFW (leaf fresh weight at tasseling), MLDW (leaf dry weight at maturity), HSW (hundred seed weight), YH (yield per hectare), MRFW (root fresh weight at maturity), VLFW (leaf fresh weight at vegetative stage), MSDW (shoot dry weight at maturity), VLDW (leaf dry weight at vegetative stage), VSDW (shoot dry weight at vegetative stage), MLFW (leaf fresh weight at maturity), MSFW (shoot fresh weight at maturity), TSFW (shoot fresh weight at tasseling), TRDW (root dry weight at tasseling), GDW (grain dry weight), GFW (grain fresh weight), VRFW (root fresh weight at vegetative stage), Prot (protein content), TLA (leaf area at tasseling), MLA (leaf area at maturity), TPH (plant height at tasseling), SS (soluble sugar content), TRFW (root fresh weight at tasseling), VLA (leaf area at vegetative stage), K (available potassium of soil), CAT (catalase), MDA (malondialdehyde to show lipid peroxidation), Pro (proline) under full irrigation (FI), partially deficit irrigation (PDI), and severely deficit irrigation (SDI).

the highest economic performance, with returns ranging from 1,552 to 1,771 USD under FI, 1,080 to 1,259 USD under PDI and 215 to 645 USD under SDI. These gains were recorded in plots amended with activated biochar at A1 and A2 levels, while the lowest returns were consistently observed in non-amended soil across all moisture regimes.

## FTIR analysis of maize grains

FTIR analysis revealed significant changes in peak intensities of key functional groups across maize hybrids (Fig. 6). The hydroxyl group (3,281 cm$^{-1}$) showed the highest intensity increases in hybrid Dk-2088 under partially and severely deficit irrigation (PDI and SDI), with 3.25–3.89% and 1.44–3.62% increases in A1 and A2 compared to A0. Conversely, a decline in this peak was observed in Dk-6317 under similar conditions.

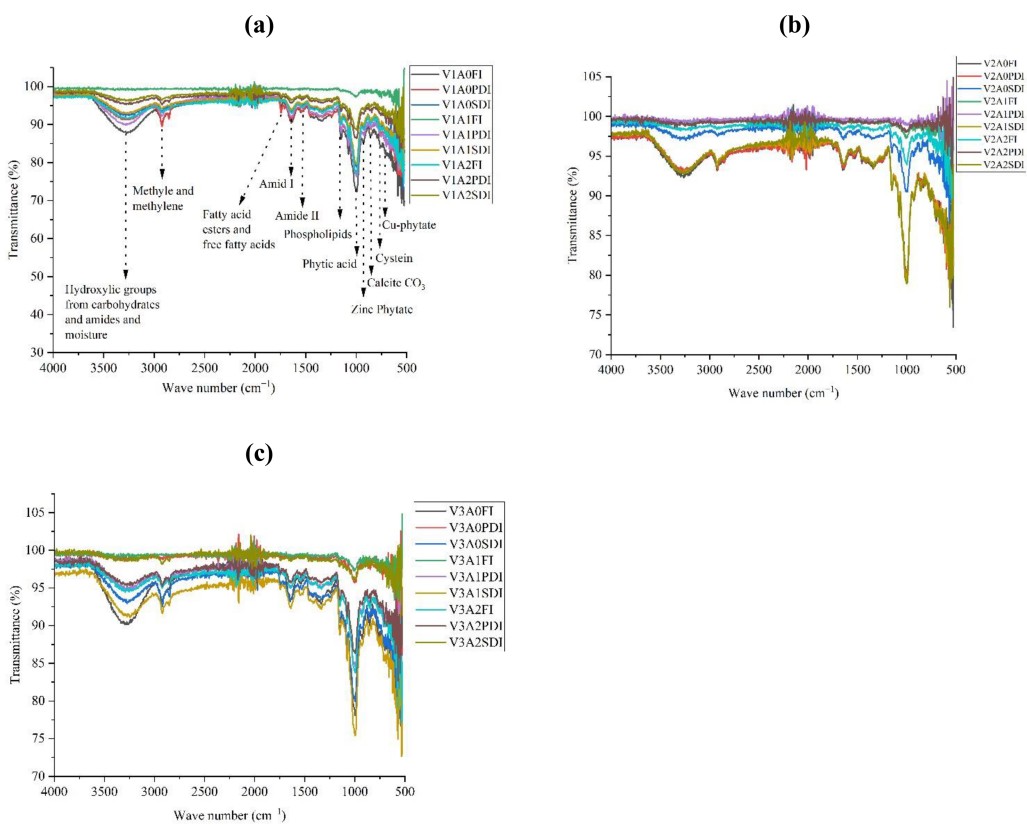

**Figure 6** **FTIR analysis of maize seeds grown in activated biochar (A) DK-2088 (B) Yh-5427 (C) DK-6321.**

The methyl/methylene group (2,926 cm$^{-1}$) exhibited marked enhancement in Yh-5427 under full and PDI conditions, particularly under A2 (up to 29%). Dk-6317 showed notable increases in fatty acid-associated peaks (1,747 cm$^{-1}$) across all irrigation regimes, peaking at 7% under SDI in A2. Amide I (1,624 cm$^{-1}$) and Amide II (1,536 cm$^{-1}$) intensities increased substantially, especially in Yh-5427 under PDI (up to 9.44%) and full irrigation (11%) respectively, while Dk-2088 showed a moderate increase under SDI. The phospholipid peak (1,152 cm$^{-1}$) increased significantly in Yh-5427 under PDI and SDI, with peak enhancements of 6.92% and 3.96% in A2, respectively. Phytic acid (1,050 cm$^{-1}$) intensity increased only in Dk-6317 under PDI and SDI, with the highest increment reaching 12.2% in A2. Meanwhile, Fe–O (584 cm$^{-1}$) intensity markedly increased in Dk-6317, reaching 24.9% under FI and A2. The MgO (786 cm$^{-1}$) and ZnO (710 cm$^{-1}$) peaks showed maximal intensities in Yh-5427 under SDI, particularly under A2 (18.2% and 25.5%, respectively). Lastly, lipid (1,755 cm$^{-1}$) and protein (1,614 cm$^{-1}$) peaks were highest in Dk-6317 and Yh-5427 respectively under PDI in A2, both exhibiting 5.7% increase compared to A0. Overall, the A2 amendment consistently led to enhanced biochemical signatures, especially under water-deficit conditions.

## DISCUSSION

Acacia wood activated biochar markedly improved soil structure and chemistry under drought conditions. The porous, high-surface-area of biochar reduced soil bulk density and increased porosity, thereby enhancing water holding capacity. The organically activated biochar at 2.5–5% (w/w) has been shown to raise water retention by 31–41% in loamy soil (*Iqbal et al., 2024*). In this study, FTIR characterization of the activated biochar revealed abundant O—H, C—H, and C=O functional groups, and SEM-EDX images showed a highly porous matrix. These features are consistent with improved soil moisture availability observed in activated biochar-amended plots. Overall, activated biochar addition significantly increased organic matter, phosphorus, and available potassium, our results are in agreement with those of *Rizwan et al. (2016)*. Such improvements are largely attributed to the inherent property of woody biochar which is typically rich in lignin, potassium and phosphorus (*Oram et al., 2014*). Previous studies have also indicated that high temperature biochar tends to contain elevated phosphorus concentrations(*Bruun et al., 2017*). Moreover, biochar improves the overall physicochemical properties of soil and enhances saturation percentage due to its internal pore network (*Ajayi & Horn, 2016*). The addition of biochar which contains high proportion of recalcitrant carbon and essential minerals, can strongly influence soil chemical attributes by promoting long term nutrient retention (*Abrishamkesh et al., 2015*; *Zhang et al., 2012*). *Shenbagavalli & Mahimairaja (2012)* reported that biochar incorporation in soil can raise soil organic matter content by 33–35% due to increase in carbon content of biochar post pyrolysis. Biochar produced at high temperatures can slightly enhance the electrical conductivity of soil because of its high mineral content, which are released upon hydration (*Joseph et al., 2021*; *Singh, Singh & Cowie, 2010*).

The observed neutralization of slightly acidic soil pH may be due to the thermal degradation of acacia wood. Its carbon content releases anions, which react with hydronium ions in the soil, thereby increasing pH (*Verheijen et al., 2009*). This liming effect is also supported by previous studies, which indicated the rise in pH following biochar addition, largely due to basic cations and ash content (*Hilioti et al., 2017*; *Mandal et al., 2018*). Moreover, anionic functional groups such as phenolic, hydroxyl, and carboxylic moieties in biochar bind hydrogen ions and increase the soil pH (*Chintala et al., 2014*).

Recent SEM-EDX analyses revealed that wood-based biochar contains both organic and inorganic minerals including Mg, C, K, S, Ca, Fe, Ti, O, and Si (*Singh et al., 2025*). Oxygen in biochar is associated with hydroxyl, phenolic, carboxylic and carbonyl groups, while hydrogen is linked with aliphatic and aromatic surface structures. During high-temperature pyrolysis, the carbon in biomass condenses into aromatic structure (*Baldock & Smernik, 2002*). The current elemental analysis also indicated an increase in carbon content in soil following the application of activated acacia biochar (*Xiao, Chen & Chen, 2016*). A slight increase in aluminum content was also possibly due to biochar mediated minerals mobilization (*Siecińska & Nosalewicz, 2017*). The FTIR analysis of biochar-amended soils showed distinct peaks at 3,600 cm$^{-1}$, corresponding to O–H stretching of dioctahedral clays like Kaolinite and polysaccharides derived from the pyrolysis of lignocellulosic materials

(*Abbas et al., 2019*; *Zeng et al., 2020*). Peaks at 1,210, 1,265, and 1,597 cm$^{-1}$ were attributed to ester C=O and hemicellulose C–O functional groups, while Si–O bands at 1,006 cm$^{-1}$ and 1,045 cm$^{-1}$ indicated the presence of phyllosilicates and quartz (*Peternele & Da Costa, 2014*).

Maize is highly sensitive to water deficiency, which induces significant biochemical stress. Severe drought leads to osmotic stress and enhanced production of reactive oxygen species (ROS) such as $O_2^-$ and OH radicals, which cause lipid peroxidation, membrane degradation, and enzyme dysfunction (*Fathi & Tari, 2016*; *Kordrostami, Rabiei & Ebadi, 2019*). To mitigate ROS damage, plants activate antioxidant enzymes like CAT and POD (*Shafiq, Akram & Ashraf, 2019*). At present higher antioxidant activity was observed in non-amended soil, particularly in DK-2088, while biochar-treated plants showed reduced activity. This aligns with earlier findings that activated biochar reduces CAT, POD, and MDA content under drought stress (*Farhangi-Abriz & Torabian, 2017*; *Shakeel et al., 2022*). Additionally, drought stress led to higher proline accumulation in non-amended plants, consistent with earlier reports linking proline synthesis to osmotic regulation under water stress (*Ozturk et al., 2021*; *Peake, Reid & Tang, 2014*). However, proline accumulation was significantly reduced in biochar-treated soils (*Shakeel et al., 2022*), suggesting improved water availability and reduced stress.

Drought is a major constraint to crop productivity, responsible for impairing photosynthesis, and delaying reproductive development (*Fahad et al., 2017*). Currently, drought stress reduced the morphological attributes of plants in non-amended soil. whereas biochar-treated plants exhibited better performance. Notably, maize grain yield increased with higher biochar doses, highlighting its potential to buffer against water stress. This improvement may be attributed to the enhanced plant available moisture and nutrient retention by wood derived biochar (*Tan et al., 2017*). Water stress during flowering stage delays cob formation and reduces yield components such as kernel rows per ear and kernel number (*Li et al., 2022*). Present findings confirm that biochar contributes to improved growth and yield, even under limited irrigation.

Optimization of biochar application rates and management practices can maximize profitability and minimize risks (*Latawiec et al., 2021*). In the present study, activated biochar-treated soils produced significantly higher grain yield (kg hectare$^{-1}$) and net profit than untreated controls. This is consistent with earlier studies reporting yield improvements due to biochar (*Sabagh et al., 2018*; *Jahan et al., 2023*). Among the tested maize hybrids, DK-6317 achieved the highest yield and profitability across all irrigation regimes. In conclusion, Acacia wood-derived activated biochar is an eco-friendly and sustainable soil amendment that increases soil fertility, improves drought resilience, and revives maize productivity.

FTIR spectra identified nutrient-rich functional groups in maize grains, including: the broad absorption region between 3,600–3,025 cm$^{-1}$ that corresponded to hydroxyl (–OH) groups from carbohydrates and amides, signifying the presence of carbon (C), hydrogen (H), oxygen (O), and nitrogen (N) (*Forato, Bernardes-Filho & Colnago, 1998*; *Naumann, Heine & Rauber, 2010*). Fatty acids and their esters were identified at 1,743 and 1,709 cm$^{-1}$, while protein structures were evident from peaks at 1,643 and 1,539 cm$^{-1}$

(*Duodu et al., 2001*). Important nutrient-rich compounds such as phospholipids and phytic acid were detected in the 1,400–800 cm$^{-1}$ region. Phytic acid, a major phosphorus storage compound, appeared at 1,050 cm$^{-1}$ and is widely recognized in food material analyses (*Lehrfeld & Morris, 1992*; *Phillippy, 2003*). Under drought conditions, peak intensities for functional groups associated with carbohydrates, proteins, and amides declined in non-amended soils, indicating nutrient loss or degradation (*Ogbaga et al., 2017*; *Mahmoud et al., 2022*). Water stress likely disrupted protein conformation and impaired chlorophyll function, resulting in reduced photosynthetic efficiency and carbohydrate synthesis. However, biochar-amended soils (A1 and A2) preserved or enhanced the intensities of these peaks, reflecting improved nutrient stability and biochemical composition. This aligns with earlier findings that biochar enhances soil carbon and nitrogen availability, supporting protein and carbohydrate synthesis even under stress (*Wang et al., 2021*; *Gharred et al., 2022*). The Yh-5427 outperformed other hybrids, showing the most consistent enhancement in nutrient peaks (*e.g.*, proteins, phospholipids, ZnO), indicating superior stress resilience and nutrient assimilation under deficit irrigation. While this one-year study demonstrates the beneficial effects of activated acacia biochar on maize growth and soil properties under different irrigation regimes, long-term field trials are essential to confirm the consistency and sustainability of these benefits across seasons and climatic variations.

## CONCLUSION

Biochar is widely recognized for its ability to improve soil water retention capacity, primarily due to its highly porous structure, which facilitates the retention of water as well as nutrients. This property ensures sustained availability of essential resources to plants during drought conditions. The increasing adoption of organic soil amendments reflects their potential in promoting climate resilient and sustainable agricultural practices. Findings from the current one-year field experiment demonstrated the detrimental effects of drought on maize development, evidenced by significant reductions in shoot, root, and leaf growth parameters across vegetative, tasseling, and maturity stages, which finally caused an adverse reduction in maize yield. However, the application of Acacia wood-derived activated biochar proved effective in alleviating drought-induced stress, as evidenced by the enhanced growth and yield attributes of maize, including increased photosynthetic pigment levels, soluble sugar content, hundred seed weight, and yield per hectare. Activated biochar plays a pivotal role in promoting sustainable agriculture by improving soil structure, rehabilitating degraded soils, and mitigating the impacts of climate change; in the present study, biochar-amended soils enhanced maize growth under drought stress by improving overall physiological performance, though long-term, multiyear studies are recommended to validate the consistency, sustainability, and adaptability of these benefits under changing environmental conditions. Organically activated acacia wood biochar represents an environmentally sustainable option that also enhances soil fertility. Further research is warranted to scale up biochar applications across diverse cropping systems and agro-ecological zones.

## ACKNOWLEDGEMENTS

We would like to acknowledge Buyers, Lahore, Pakistan, for providing seeds of maize hybrids. Dr. Fahd Rasul, Associate Professor, University of Agriculture, Faisalabad for helping in biochar preparation.

### Funding
This work was financially supported by the Higher Education Commission (HEC), of Pakistan under NRPU grant No. 20-16716. The funders had no role in study design, data collection and analysis, decision to publish, or preparation of the manuscript.

### Grant Disclosures
The following grant information was disclosed by the authors:
Higher Education Commission (HEC), of Pakistan: 20-16716.

### Competing Interests
The authors declare there are no competing interests.

### Author Contributions
- Zarghoona Naz conceived and designed the experiments, performed the experiments, analyzed the data, prepared figures and/or tables, and approved the final draft.
- Audil Rashid analyzed the data, authored or reviewed drafts of the article, and approved the final draft.
- Summera Jahan conceived and designed the experiments, performed the experiments, analyzed the data, prepared figures and/or tables, and approved the final draft.

### Data Availability
The data are available in the Supplementary File.

### Supplemental Information
Supplemental information for this article can be found online at http://dx.doi.org/10.7717/peerj.20048#supplemental-information.

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
