# Peer review of "Interactive effects of acacia biochar, maize hybrids, and irrigation levels on soil health and crop productivity"

_PeerJ, doi:10.7717/peerj.20048_

## Round 0.1 · original submission · Major Revisions

Dear Authors
The manuscript cannot be accepted for publication in its current form. It needs a major revision before publication. The authors are invited to revise the paper, considering all the suggestions made by the reviewers. Please note that the requested changes are required for publication.
With Thanks

Reviewer 1 ·

Basic reporting

The study highlights the potential of activated acacia biochar in improving soil health, water retention, and nutrient availability, leading to enhanced maize growth and yield under varying moisture conditions. The research could be a significant addition to the journal. However, some queries need to be answered for the improvement of this manuscript.

The term "activated biochar" should be consistently used throughout the manuscript to maintain the clarity and technical accuracy.
The rationale for studying various physicochemical attributes needs to be incorporated in the introduction section.
Describe the novelty of the research work in the introduction section.
Check the grammar throughout the manuscript.

Experimental design

Describe the method(s) used for biochar activation (e.g., chemical, physical or thermal activation), along with reference.
Add suitable reference(s) for catalase estimation, economic analysis and statistical analyses.

Validity of the findings

Explain how various physio-biochemical and growth attributes provide insights into the soil-plant interactions influenced by activated biochar.
In discussion section include references related to the economic analysis of yields to support the feasibility and cost-effectiveness of using activated biochar in agricultural systems.
The reasons behind the decrease in antioxidants should be clearly explained.
Discuss the possible interactions between biochar amendments and plant physiological responses leading to changes in antioxidant levels.
A more in-depth analysis of the results should be included, discussing the broader implications of biochar applications on soil and plant health.
The policy implications of the present research should be discussed, particularly regarding sustainable agriculture and soil fertility management.
In conclusion section, future directions may be suggested, including scaling up biochar applications and assessing its long-term environmental impacts.

Reviewer 2 ·

Basic reporting

The manuscript provides a valuable investigation into the impact of biochar on maize performance; however, several aspects require clarification for better transparency and understanding. Specifically, the methods and results sections should provide more details on the use of vermicompost and perlite-based biochar, as these materials are central to the study but are not adequately described. Additionally, more information on the experimental design, including soil conditions, maize hybrid selection, and water stress impact, is necessary to better contextualize the findings

Experimental design

The experimental design should include detailed descriptions of the materials used, particularly the vermicompost and perlite-based biochar. It is important to provide information on their preparation methods, composition, and application rates to ensure reproducibility of the study. Additionally, the experimental conditions, including soil type, irrigation practices, and other environmental factors, should be clearly described. This will allow readers to better understand the context in which the experiment was conducted and how these factors might influence the results.

Validity of the findings

A more thorough explanation of the experimental conditions—such as soil type, irrigation practices, plant and soil nutrients —will help contextualize the findings and demonstrate how these factors may have influenced the results

Additional comments

The study addresses an important topic, exploring the potential role of biochar in improving crop yield and water use efficiency, which is crucial for sustainable agriculture, especially in the face of climate change. The specific objectives of the study are to assess the impact of biochar-enriched soil on maize yield under varying drought conditions. Specifically, it aims to investigate the effectiveness of vermicompost and perlite-based activated biochar in improving maize performance in field conditions.
Specific comments
1. Line 95-96: The manuscript mentions investigating the efficacy of vermicompost and perlite-based activated biochar in improving maize performance. However, the Methods and Results sections do not provide sufficient details regarding the use of vermicompost or perlite-based biochar. It is important to describe how these materials were prepared, applied, and evaluated, as well as their specific roles in the study. Including this information will clarify the experimental setup and improve the transparency of the findings.
2. Line 110: “The samples from biochar amended soil were collected to analyze the physico-chemical status of the soil”, Specify the duration of the experiment to provide context for soil sample collection. Indicate whether the samples were taken at multiple time points or at the end of the experiment to assess changes over time.
3. Provide more details on the field conditions, including soil type, climate, temperature, moisture levels, and any other relevant environmental factors. This information is essential for reproducibility and understanding the experiment’s context.
4. The manuscript lacks data on plant nutrient uptake, which is essential for understanding the impact of biochar on plant nutrition. Since biochar can influence soil nutrient availability and uptake efficiency, including this information would provide a more comprehensive assessment of its role in improving plant growth and yield. Consider adding relevant data on macronutrient and micronutrient uptake, as well as discussing potential mechanisms linking biochar amendments to enhanced nutrient assimilation and yield improvement.
5. f interaction analysis has been conducted, consider including nutrient interactions as well. Since biochar enhances soil nutrient availability, examining how nutrients interact with soil properties and plant growth characteristics would provide deeper insights into its effects on plant nutrition and yield. Additionally, assessing how drought conditions influence these interactions is crucial, as water availability can significantly affect nutrient uptake and biochar's effectiveness in improving plant resilience.
6. The plot size (4×4 m²) is relatively small, which may raise concerns about the scalability of the findings. Please clarify how economic analyses were conducted in relation to this limited plot size. Were the results extrapolated to larger field conditions, and if so, what assumptions were made? Additionally, specify the economic parameters considered (e.g., cost of biochar application, labor, yield increase, return on investment) and how they were assessed to ensure the analysis reflects realistic agricultural conditions.
7. The conclusion is too general and lacks specific mechanisms of water stress tolerance. It would be beneficial to describe how biochar contributes to improving water stress tolerance in plants, particularly through its influence on soil moisture retention and nutrient availability. Additionally, the linkage to plant nutrition is not clearly established. Explain how plants can accumulate nutrients from biochar, including any physiological or biochemical processes involved. Consider discussing the soil biological and chemical changes that occur after biochar application, particularly how microbial activity, nutrient cycling, and enzyme activity are impacted. Focusing on enzymes related to soil nutrient availability would also provide valuable insights into the biochar's role in enhancing soil fertility and plant growth under stress conditions.
8. It would be helpful to explain why three maize hybrids were selected for the study. Are these hybrids genetically different or adapted to different environmental conditions, such as drought tolerance? Clarifying the rationale for selecting these specific hybrids and how they differ from each other in terms of growth characteristics, nutrient requirements, or stress resilience will strengthen the study's design and provide a clearer context for the results.

Reviewer 3 ·

Basic reporting

In the present study entitled "Bridging growth and sustainability: biochar's impact on yield and water dynamics in diverse maize hybrids under natural field conditions", authors found that maize productivity was enhanced by the activated acacia biochar under water-limited conditions. I think that the work falls into the scope of the journal and the findings are interesting, however MS demands major revision.

Abstract: The abstract can be more concise. I would suggest adding numerosity associated with each parameter for a better understanding of the main outcomes and context of the research. abstract. Add/change some keywords.

Introduction: Authors should add the novelty of the research and conceive a strong idea from the review literature about limitations and Challenges to coping with research gaps to conceive strong idea from literature review highlighting a) Standardization of biochar properties b) long-term impacts on soil properties need more investigation while more studies focus on short-term impact on soil water retention c) little focus on physiological studies etc.

Experimental design

Materials and methods: How many replications per treatment? How many plants per replication? Day/light hrs? Temperature? The drafting of many sentences needs to be revised. Please standardize hour for hour/hours. Materials and methods require more information with proper instrument name and model. No doubt, the author provided some details among various sections, but I think it is better to check all sections.

Validity of the findings

Results and Discussion: In results, there is a connection between sentences and paragraphs, but I suggest restructuring the text to avoid making contradictory statements between results and discussion. There are many values in results that increase the ubiquity of results. I would suggest presenting your results by increase/decrease %age. The percentage should be up to two digits e.g., 8% instead of 8.01%. One way of improving Discussion is to avoid repetition of results in this part. In discussion, there is a lack of a mechanistic approach. Spellings and the English language need to be checked thoroughly. Overall, the drafting of many sentences needs to be improved. Tidying up the text is also suggested.
The conclusions should be supported by data on how they are linked to goals and how this information contributes to knowing the gaps.
Tables. The homogenous group should be added as a superscript with the mean value and SE mean to know if there is a significant difference among the treatments or not.
Figures. I noticed many meteorological parameter graphs; in my opinion, all parameters should be combined into a single plot and coloured by line or graph.

---

## Round 0.2 · Minor Revisions

Dear Authors

The manuscript still needs a minor revision before publication. The authors are invited to revise the paper, taking into account all the suggestions made by the reviewer.

Moreover, given that this study was conducted over only one growing season, the results do not fully reflect the long-term effects of biochar on soil properties and crop performance. Therefore, I recommend that the authors clearly indicate that these findings should be considered preliminary. Additionally, the manuscript should include a statement outlining plans for long-term experiments to validate and expand upon these initial observations.

Please note that the requested changes are required for publication.

With Thanks

Reviewer 1 ·

Basic reporting

The authors have satisfactorily addressed all the concerns raised by the reviewers. The revised manuscript represents significant improvement, particularly in terms of grammar and clarity of context. The background of study is now supported by sufficient and relevant literature.

Experimental design

The experimental design is well structured and supported with appropriate references. The detailed methodology is adequate to allow replication of the experiments, strengthening the authenticity of the study.

Validity of the findings

The research findings are well justified using relevant and recent literature. Conclusions have been stated well and linked to the research question.

Reviewer 2 ·

Basic reporting

The title of the manuscript, “Bridging growth and sustainability: biochar's impact on yield and water dynamics in diverse maize hybrids under natural field conditions,” is too broad relative to the actual scope of the study. I recommend specifying the title to better reflect the main focus and experimental details.
Moreover, field trials intended to assess the effects of biochar on crop yield and water dynamics generally require at least three years of replicated experiments to account for inter-annual variability in weather and soil conditions. Conducting the study over a single season makes it difficult to draw robust conclusions about the consistency and reliability of biochar’s effects. I recommend acknowledging this limitation explicitly in the discussion and, if possible, planning multi-year trials in future work to strengthen the evidence base.

Experimental design

The study was conducted over a single growing season. However, assessing the effects of biochar typically requires longer-term observation, as its impact on soil properties, water dynamics, and crop performance often develops gradually over multiple years. Therefore, the conclusions drawn about biochar’s effectiveness may not fully reflect its long-term influence. I recommend that the authors clearly state this limitation in the manuscript and interpret the results with appropriate caution

Validity of the findings

The conclusion should be more specific and directly aligned with the stated aim and findings of the study. General statements such as “Undoubtedly, activated biochar plays a pivotal role in fostering sustainable agricultural production, rehabilitating barren soils, and mitigating climate change effects” are too broad and not fully supported by the one-year field data presented. I recommend revising the conclusion to focus on the observed effects of organically activated acacia wood biochar on maize yield and water dynamics under the specific experimental conditions

Reviewer 3 ·

Basic reporting

Title
The title clearly describes the article.

Abstract
The abstract is well structured.

Introduction
The introduction is up to mark of scientific background and highlights the updated literature review thus representing aims and objectives.

Experimental design

Materials and Methods
The methodology is well written and all components are clearly described now.

Validity of the findings

The author significantly improved a large part of the language to ensure meaningful sentences.

Additional comments

The authors responded positively to most of the comments.

---

## Round 0.3 · accepted · Accept

Dear Authors,

I am pleased to inform you that the manuscript has been improved following the last revision and can now be accepted for publication.

Congratulations on accepting your manuscript. Thank you for your interest in submitting your work to PeerJ.

With Thanks